# Evolution and Functional Dynamics of TCP Transcription Factor Gene Family in Passion Fruit (*Passiflora edulis*)

**DOI:** 10.3390/plants13182568

**Published:** 2024-09-13

**Authors:** Munsif Ali Shad, Songguo Wu, Muhammad Junaid Rao, Xiaoying Luo, Xiaojin Huang, Yuxin Wu, Yuhong Zhou, Lingqiang Wang, Chongjian Ma, Lihua Hu

**Affiliations:** 1Henry Fok School of Biology and Agriculture, Shaoguan University, Shaoguan 512005, China; p2022027@gxu.edu.cn (M.A.S.); lqwang@gxu.edu.cn (L.W.); 2State Key Laboratory for Conservation and Utilization of Subtropical Agro-Bioresources, Guangxi University, Nanning 530004, China; wusongguo1128@163.com (S.W.); lxy2090151433@163.com (X.L.);; 3State Key Loboratory of Subtropical Silviculture, College of Forestry and Biotechnology, Zhejiang Agriculture and Forestry University, Hangzhou 311300, China; mjr@zafu.edu.cn; 4College of Life Sciences and Technology, Huazhong University of Sciences and Technology, Wuhan 430074, China; u202213718@hust.edu.cn

**Keywords:** passion fruit, TCP transcription factors, cold stress, GFP subcellular localizations, miRNA319s

## Abstract

Passion fruit is a valued tropical fruit crop that faces environment-related growth strains. *TCP* genes are important for both growth modulation and stress prevention in plants. Herein, we systematically analyzed the *TCP* gene family in passion fruit, recognizing 30 members. Genes exhibiting closer phylogenetic relationships exhibited similar protein and gene structures. Gene members of the *TCP* family showed developmental-stage- or tissue-specific expression profiles during the passion fruit life cycle. Transcriptome data also demonstrated that many *PeTCPs* showed induced expression in response to hormonal treatments and cold, heat, and salt stress. Based on transcriptomics data, eight candidate genes were chosen for preferential gene expression confirmation under cold stress conditions. The qRT-PCR assays suggested *PeTCP15/16*/*17*/*19*/*23* upregulation, while *PeTCP1*/*11*/*25* downregulation after cold stress. Additionally, *TCP19*/*20*/*29*/*30* exhibited in silico binding with cold-stress-related miRNA319s. GFP subcellular localization assays exhibited PeTCP19/1 were localized at the nucleus. This study will aid in the establishment of novel germplasm, as well as the further investigation of the roles of *PeTCPs* and their cold stress resistance characteristics.

## 1. Introduction

Since passion fruit (*Passiflora edulis*) has significant edible, medicinal, and ornamental value, it is widely grown in tropical and subtropical regions worldwide. Many consumers relish its egg-shaped fruit due to its unique flavor, rich aroma, acid pulp, and yellow juice [1]. Owing to the abundance of alkaloids, flavonoids, and other physiologically active components found in passion fruit, extracts derived from its leaves, fruits, peels, and seeds have therapeutic properties that include anti-inflammatory, soothing, antioxidant, and anticancer characteristics [2]. Studies suggest that passion fruit is an effective resource for managing rocky desertification and fostering industrial growth [3]. Passion fruit peels (PFPs) are increasingly being recognized as valuable waste materials that can serve as feedstock for pectin extraction [4]. Because most passion fruit varieties have large floral organs, bright coronal filaments, a rich fragrance, and lush branches and leaves, they are used as ornamental plants for flower racks due to their ornamental value [5]. Passion fruit has a huge agricultural significance as it has a short growth period, can grow in a variety of soils, uses less soil surface due to vertical climbing habits, has 5–7-year semi-perennial vines, is easy to propagate from stem cuttings, and is adapted to subtropical and tropical regions [6]. Identifying and characterizing important gene families in passion fruit might help boost the growth of the world’s agricultural economy.

Transcription factors (TFs) are major elements of the genetic foundation for phenotypic evolution. TCP proteins (TCPs) are a class of plant-specific TFs, that was first identified and designated as *TEOSINTE BRANCHED 1* (*TB1*) in *Zea mays* [7], *PROLIFERATING CELL FACTORS 1* and *2* (*PCF1* and *PCF2*) in rice [8], and *CYCLOIDEA* (*CYC*) in *Antirrhinum majus* [9]. TCP domains are defined by an atypical 59-amino acid basic helix–loop–helix motif structural feature and are unrelated to the DNA-binding bHLH domain. Based on TCP domains, this class of protein is partitioned into Class I (TCP-P or PCF type) and Class II (TCP-C type) [10]. Class I is distinguished by a four-amino-acid deletion which is a conserved feature. Class II is divided into the CIN and CYC clades [11]. Two duplication events in the CYC clade among the main eudicots led to three subgroups designated as CYC1, 2, and 3 [12]. The accumulated research evidence suggested that *TCP* genes were implicated in many growth-related mechanisms including axillary meristem development, flower and leaf morphology, circadian rhythm regulation, hormone signaling, seed germination, and defense [13,14].

Plants constantly face adverse environmental conditions, and researchers have identified a correlation between TCPs (TEOSINTE BRANCHED 1, CYCLOIDEA, and PCF) and plant responses to abiotic stresses [15]. Plant growth and development are significantly constrained by cold stress [16]. Plants cope with cold stress through modification such as promptly increasing the concentration of calcium ions, a saturation of cytosol affecting plant hormones such as gibberellin (GA), abscisic acid (ABA), and brassinolide (BR), arresting the growth, leading to lipid peroxidation and fatty acid unsaturation, modifications in phospholipid configurations, and accumulation of osmoprotectants like soluble sugars, proline, and betaine [16]. Knockdown of two rice *TCP* genes *OsPCF5* and *OsPCF8* resulted in enhanced tolerance to cold stress after chilling treatment [17]. Notably, two *OsTCPs*, *OsPCF6* and *OsTCP21*, demonstrated significant cold-induced expression, with knockdown plants exhibiting enhanced cold tolerance compared to wild-type plants due to improved reactive oxygen species scavenging [18]. *PCF6* expression in sugarcane seedlings exposed to cold stress for 24 h was reduced to 50 percent [19]. Similarly, the cassava *TCP* gene *MeTCP3a* and *MeTCP4* expressions were reduced after their seedlings were treated with 4 °C cold stress [20]. In transgenic creeping bentgrass four members of the TCP gene family, namely *AsPCF5/6*/*8*/*14*, exhibited expressional depression in conjunction with increased salt and drought tolerance linked with enhanced water retention and leaf wax contents [21]. Contrarily, overexpression of the rice *OsTCP19* resulted in abnormal development including reduced formation of lateral roots and enhanced abiotic stress tolerance [22]. *ZmTCP14* overexpression under drought conditions led to a significant reduction of drought tolerance, while gene-edited lines of *ZmTCP14* demonstrated enhanced drought tolerance, suggesting it acts as a negative regulator of drought stress [23]. *TCP10* from *Moso bamboo* exhibited induced expressions under drought stress and its overexpression in rice and *Arabidopsis* enhanced drought tolerance in transgenic plants [24].

Owing to the development of genome sequencing technologies, *TCP* gene families have been studied in various modal organisms of agro-botanical importance like cultivated rye [25], switchgrass [26], potato [27], and *Arabidopsis* [28]. Recently, the genome sequence of passion fruit was publicly released [20,29,30], offering valuable genome resources for the identification of novel genes. Despite the agricultural importance of passion fruit, there is a significant lack of information regarding the comprehensive characterization and functional analysis of the TCP transcription factor gene family in this plant. This study aimed to address this gap by achieving the following objectives: (1) Systematic identification of passion fruit *TCP* genes. (2) Description of the conserved domains and *cis*-regulatory elements in their sequences. (3) Exploration of the distribution of *TCP* genes in the passion fruit genome. (4) Analysis of the evolutionary relationships among these genes to understand their origins and divergence between different subgroups. (5) Expression profiling of *TCP* genes in various sugarcane tissues, developmental stages, and under different stresses. (6) Expressional validation of a few promising candidate genes under cold stress. (7) Confirmation of TCP proteins’ subcellular localization through GFP assays.

This study provides valuable insights into the sequence characteristics, genomic distribution, evolutionary background, and functionality of *TCP* genes in passion fruit. This study will serve as raw materials for further molecular characterization and stress breeding of passion fruit employing TCP transcription factors.

## 2. Material and Methods

### 2.1. Identification and Sequence Analysis of Passion Fruit TCP Proteins

Firstly, TCP protein sequences were retrieved from the yellow passion fruit genome in the Passion Fruit Genomic Networks database (http://passionfruit.com.cn/cgi-bin/motif_expression.pl (accessed on 4 August 2023)) using the TCP PFAM accession ID (PF03634). Later on, two published genomic assemblies named GWHAZTM00000000 and GWHANWG00000000 of purple passion fruit types were downloaded from Genome Warehouse (https://ngdc.cncb.ac.cn/gwh/Genome/557/show (accessed on 4 August 2023)). Similarly, the latest whole-genome protein sequences of *Arabidopsis* from the Phytozome database (https://phytozome-next.jgi.doe.gov/ (accessed on 4 August 2023)) were downloaded and uploaded to Bioedit software 7.2 (accessed on 10 August 2023) to generate three local genome files. The yellow passion fruit TCP protein sequences were used as queries to perform a local BLASTP against the proteomes in Bioedit software with rigorous thresholds of E value < 1 × 10^−5^, query cover > 50%, and protein identity > 30%. The Online Conserved Domains search tool in the NCBI database (https://www.ncbi.nlm.nih.gov/Structure/cdd/wrpsb.cgi (accessed on 19 August 2023)) was utilized to confirm the protein sequences with the characteristics of bHLH domains. The physicochemical properties of yellow passion fruit TCP proteins were computed using Expasy ProtParam (https://web.expasy.org/protparam/ (accessed on 22 September 2023)). The Euk-mPLoc (http://www.csbio.sjtu.edu.cn/bioinf/euk-multi/ (accessed on 25 September 2023)) tool was used to predict the subcellular localization of all putative TCPs.

### 2.2. Multiple Sequence Alignment and Phylogenetic Tree Construction

Using MEGA-11 software (accessed on 4 September 2023) [31], the full-length protein sequences of *Arabidopsis* (24) and PeTCPs (30) were aligned to construct a neighbor-joining phylogenetic tree. The criteria were adopted using 1000 bootstrap replications of the Poisson correction model and a pairwise deletion option. Gene structures of the identified *PeTCPs* were determined by the Gene Structure Display Server 2.0 tool (http://gsds.gao-lab.org/ (accessed on 9 October 2023)). Domain organizations of the likely PeTCPs were initially identified using the NCBI Batch CDD server (https://www.ncbi.nlm.nih.gov/Structure/bwrpsb/bwrpsb.cgi (accessed on 10 October 2023)) and, later on, depicted through TBtools software (version 2.019) accessed on 11 October 2023) [32]. Using the TBtools program, the arrangement of *PebTCPs* throughout the nine chromosomes of the *P. edulis* genome as well as the duplications was mapped, and Ka/Ks values were computed.

### 2.3. Structural Analysis of TCP Proteins

Homology modeling of PeTCP proteins was performed and a single representative member structure of three subfamilies was modeled based on the SWISS-MODEL database in user-specified template mode. The Arabidopsis TCP proteins representing each subfamily AlphaFold structure were downloaded from UniProt and used as a template. Then these templates were used in SWISS-MODEL (https://swissmodel.expasy.org/ (accessed on 19 January 2024) ) to model the characteristic DNA-binding 59-amino-acid bHLH domains of PeTCPs. Eventually, the modeled domains were visualized and superimposed in Chimera 1.15 software (accessed on 25 January 2024) to observe the structural differences [33].

### 2.4. Cis-Regulatory Element Analysis of the Promoter Regions of TCP Genes

The yellow passion fruit genome was used in this study, therefore the full genomic sequence of yellow passion fruit was downloaded from the Passionfruit Genomics Network website (http://passionfruit.com.cn/search_downloads.html (accessed on 6 February 2024)). The full genome was viewed in UltraEdit software (https://www.ultraedit.com/ (accessed on 6 February 2024)). The first 20 bp of the first starting codon of each *PeTCP* coding sequence was searched manually using Ctrl+F keyboard functions. Following matching, the upstream 2000 bp promoter sequences were collected. Using genome browser functions on the Passionfruit Genomics Network website, the upstream gene cds sequences were also examined to partition the upstream gene CDS and under-query gene promoters. The promoter sequences were then fed to the PlantCARE web tool [34]. Subsequently, overlapping sites were removed manually in MS Excel software (v2016) (accessed on 10 February 2024) and the final representations were created through Tbtools software [32].

### 2.5. In Silico Binding Prediction between TCP Genes and miRNA319

Rice microRNA319 positively regulates cold stress by repressing the expression of its target *OsTCPs*, and since microRNA sequences are conserved among plant species, we tested whether OsmiRNA319 could target *PeTCPs*. Rice miR319, miR319a, and miR319b sequences were downloaded from the plant microRNA database (PMRD) (http://bioinformatics.cau.edu.cn/PMRD (accessed on 14 March 2024)) and, later on, were analyzed with *PeTCPs* CDS sequences as target genes in psRNATarget (https://www.zhaolab.org/psRNATarget/ (accessed on 16 March 2024)).

### 2.6. Expression Profile Analysis Based on RNA-Seq Data

RNA-seq datasets of all the putative *PeTCPs* were downloaded from the Passionfruit Genomics Network database (http://passionfruit.com.cn/cgi-bin/gene_expression.pl?name=Pe2g01742 (accessed on 6 April 2024)) for eight tissue samples (leaf, stem, root, petal, stamen, pistil, mature and immature fruit), abiotic stresses (heat, cold, salt, and drought), and hormonal treatments (ABA, Eth, Auxin, GA, and MeJA) with three biological replicates. For cold and heat stress treatments, 4 °C and 42 °C temperatures were used, respectively, while 25 °C was used as a control. The fragments per kilobase of transcript per million fragments (FPKM) gene expression values were used to generate heatmaps using the Tbtools software.

### 2.7. GO Annotation and KEGG Analysis of PeTCP Genes

The GO annotation and KEGG analysis of identified *PeTCPs* were performed through the Passionfruit Genomic Networks database and graphical depictions were made through the https://bioinformatics.com.cn/ (accessed on 16 April 2024) web tool.

### 2.8. Functional Validation of the Candidate PeTCP Genes through qRT-PCR

qRT-PCR assays were used to study the *PeTCP* gene expression changes under cold stress. Green healthy stem cuttings having two buds of the same length were made from yellow passion fruit vines growing in the Guangxi University vineyard. The leaves attached to the cutting were cut in half. The stem cuttings were gently inserted into pods containing compost–vermiculite mix with one bud inside the soil mix and the other outside. The roots sproute, d from the bud underneath while leaves and shoots emerged from the aerial bud. Seedlings were grown at 30 °C in a light room using 16 h light and 8 h dark periods. Cold stress treatments were applied to the two-month-old healthy passion fruit seedlings with fully developed roots and shoots. For each cold stress treatment, 8–10 seedlings were used. To apply cold stress, healthy plants were put in a growth chamber with temperatures set at 6 °C. At least three leaves (top, middle, and bottom) from each treated plant were collected at 0, 6, 12, and 24 h intervals, respectively. After stress treatments, the leaves were frozen and the RNAs of all samples were extracted employing a Vazyme RNA isolation kit, Nanjing, Jiangsu, China. An Aidlab Truescript 1st Strand cDNA Synthesis Kit (Aidlab Biotechnologies, Ltd., Beijing, China) was used to synthesize cDNA from extracted RNA. Real-time PCR was performed using the PC59-2 × SYBR Green qPCR Mix (Aidlab Biotechnologies, Ltd., Beijing, China) in the DLAB accurate96 system and primers are listed in Appendix A. The qRT-PCR conditions were 95 °C for 2 min; 40 cycles of 95 °C for 15 s, 60 °C for 30 s; and 72 °C for 30 s. Three technical replicates from three biological replicates were used for each analysis, and the 2^–∆∆Ct^ method was used to determine the fold change of each gene. For normalization of mRNA levels, the passion fruit *PeEF1a* gene was used as an internal control [35].

### 2.9. Subcellular Localization of Representative TCP Proteins

The GFP subcellular localization of PeTCP19 and PeTCP1 proteins was performed according to the methods described in Xiaojin Huang et al., 2024 [36]. The primers used for cloning of coding sequences of *PeTCP* genes are listed in Appendix A.

## 3. Results

### 3.1. Identification and Phylogeny of TCP Proteins in Passion Fruit

A total of 30 *PeTCP* genes that encode TCP proteins were identified in the yellow passion fruit genome. The genes were renamed, from *PeTCP1* to *PeTCP30*, based on their ascending order of chromosomal locations (Table 1). Furthermore, 10 and 24 *PeTCP* genes were identified from GWHAZTM00000000 (purple type) and GWHANWG00000000 genome assemblies, respectively (Appendix A). To study how all 30 identified TCP proteins are related, a neighbor-joining phylogenetic tree was constructed along with 24 TCPs of *Arabidopsis* (Figure 1). Passion fruit TCP members were classified into two classes and three subfamilies. Among passion fruit TCP proteins 15 were Class I members of the PCF subfamily. The rest of the fifteen PeTCPs of Class II were distributed into CIN and CYC subfamilies having eight and seven proteins, respectively.

Additionally, we calculated the physiochemical properties of 30 TCP proteins (Table 1). The amino acids in the proteins encoded by the 30 TCP genes ranged from 164 (PeTCP1) to 759 (PeTCP16), while their molecular weights (MWs) varied from 17,985.36 Da (PeTCP1) to 83,909.83 Da (PeTCP16). Protein isoelectric points (pI) were predicted, ranging from 5.61 (PeTCP6) to 8.86 (PeTCP10). Proteins’ thermal stability varied little among TCP proteins, with the aliphatic amino index (A.I.) revealing it ranged from 52.83 (PeTCP28) to 86.96 (PeTCP16). All TCP proteins exhibited a negative grand average of hydropathicity score (GRAVY), suggesting they are primarily hydrophilic. Finally, the subcellular localization prediction analysis revealed that except for membrane-localized PeTCP6/15/16 and cytoplasm-localized PeTCP3/13/23/26, the rest of the TCP proteins were predicted to be localized in the nucleus.

After ascertaining the phylogenetic classification and distribution of TCP proteins in the three abovementioned subfamilies we performed the multiple sequence alignment (MSA) of bHLH domains in MEGA-11 software and depicted it through the ESPript online server (Figure 2). The results indicated that the four characteristic amino acid deletions in the basic region led to the diversification of PeTCPs into two major phylogenetic classes.

### 3.2. Gene Structural Analysis and Domain Organization of PeTCP Genes

The *TCP* gene family of passion fruit was examined for its conserved domain composition and gene structure features, and their phylogenetic connections are exhibited (Figure 3A–C). The domain analysis revealed all the TCP proteins possess only TCP domains, however, two members of Class I, PeTCP22, and PeTCP16, contained additional domains called SKN1 and ACT, respectively (Figure 3A). Gene structure organization depicted that *PeTCP6* had the highest number of exons (four), whereas five genes of the CYC subfamily *PeTCP4*/*9*/*10*/*18*/*24* possessed two exons each (Figure 3B). The rest of the *PeTCPs* contained only a single exon. Gene structures and domain composition of *PeTCP* genes clustered in the same subclasses are comparable, suggesting a strong correlation between gene structures and evolutionary relationships among *PeTCPs*.

### 3.3. Homology Modeling and 3D Structural Comparisons of PeTCP Proteins

Structural examination of proteins has significant effects on comprehending their functions. We performed the homology modeling of PeTCP9 from CYC, PeTCP16 from CIN, and PeTCP25 from PCF subfamilies (Figure 4). Each subfamily protein was made up of single chains, consisting of a typical DNA-binding 59-amino-acid basic helix–loop–helix motif. The first helix was small with three turns and highly similar in all three representative proteins along with the loop region, suggesting this region is highly conserved. Contrarily, the 2nd helix exhibited structural differences such as PeTCP9 (CYC) having the longest alpha-helix with seven turns while PeTCP25 (PCF) and PeTCP17 (CIN) possessed six and five turns, respectively. These differences in 2nd helix length might be attributed to the functional variation of these proteins either through homodimerization with other TCP proteins or DNA binding. Taken together, these proteins’ homology models offer a foundational framework for delving deeper into the molecular roles of TCP proteins.

### 3.4. Chromosomal Distributions and Gene Duplication Analysis of PeTCPs

In principle, different gene duplication patterns are assumed to be the driving force behind gene family formation and the evolution of species. Among all nine passion fruit linkage groups, the *PeTCPs* were unevenly distributed (Figure 5). Chromosome 2 possessed the highest (10) number of *PeTCPs*, followed by chromosome 8 which contained 6 genes. Chromosomes 3, 5, 6, and 9 each had three *PeTCPs*, while chromosomes 1 and 4 each possessed a single gene. Duplicated genes were determined through reciprocal BLAST approaches. The location of duplicated genes on different chromosomes implied that *PeTCPs* might have arisen majorly through segmental gene duplications. Substitution ratio Ka/Ks has the functionality to describe evolutionary processes and the nature of selection or selection pressure, therefore we estimated the Ka/Ks ratios for all duplicated *PeTCP* gene pairs (Appendix A). Ka/Ks estimates indicated that the values of all duplicated gene pairs were less than 1, indicating that the *TCP* gene family might have experienced purifying selection during evolution. It is possible that the purifying selection was crucial in preserving the *TCP* genes’ conserved structure over time.

### 3.5. Prediction of Cis-Regulatory Elements in PeTCP Promoters

Probable roles of *PeTCP* genes in phytohormone responses, growth and development, and abiotic stresses were examined by analyzing the *cis*-regulatory elements (CREs) inside their promoter regions (Figure 6). A total of 291 CREs were estimated in 16 *PeTCP* genes. The predicted 291 CREs could be distributed into 134, 94, and 63 related to growth, hormones, and stress, respectively. Among all the genes, *PeTCP15* had the most CREs (25), whereas *PeTCP22* possessed the fewest with 13 CREs. Abscisic acid (ABA) responsiveness (ABRE, 33, 35.0%) and MeJA responsiveness (CGGTA-motif and TGACG-motif, 27, 28.0%) were relatively abundant among CREs related to hormonal responsiveness in the promoter regions of *PeTCPs* (Figure 6A). On the other hand, CREs associated with salicylic acid (SA) responsiveness (16, 17.0%), gibberellin (GA) responsiveness (12, 12.7%), and CREs linked to auxin (IAA) responsiveness (6, 6.8%) were sporadically distributed. CREs involved in regulating growth and development (Figure 6B), such as CAT-box for meristem expression (10, 7.4%), GCN4_motif for endosperm expression (1, 0.7%), RY-element for seed-specific regulation (1, 0.7%), and circadian for circadian control (3, 2.2%), were relatively insufficient compared to the light-responsive elements (116, 86.5%). Regarding CREs associated with stress responsiveness (Figure 6C), the anaerobic induction elements (AREs) (38, 60.3%) were the most prevalent across all genes. Conversely, low-temperature responsiveness (LTR) (9, 14.0%), defense, stress responsiveness (10, 15.0%), and drought responsiveness (6, 9.5%) CREs were unevenly distributed among different genes. These results reveal significant diversity in the composition and quantity of CREs in the promoter regions of *PeTCPs*, suggesting that different CREs regulate *TCP* gene expression in passion fruit.

### 3.6. Prediction of Putative TCP Genes Targeted by miRNA319

We tested the in silico binding of miRNA319, a conserved class of plant cold-stress-related microRNAs with *PeTCP* genes. Our analysis indicated that miRNA319a and miRNA319b can bind with *TCP19/20/29/30* genes (Figure 7). Interestingly all four genes were phylogenetically conserved, belonging to the CIN subfamily of Class II.

### 3.7. Expression and GO/KEGG Enrichment Analysis of TCP Genes

Transcriptomic data were used to characterize the expression profiles of the *PeTCP* genes at different developmental stages to investigate the potential functions of these genes (Figure 8A, Appendix A). The hierarchical clustering of expression patterns allowed the *PeTCP* genes to be sorted into different groups. Different groups of *PeTCP* genes had unique patterns of temporal and spatial expressions. The heatmap clustering indicated that *PeTCP17*/*23* in petals and stamens, *PeTCP11*/*29* in root, *PeTCP19*/*29* in leaf, and *PeTCP1*/*15* in immature fruit tissue exhibited high preferential gene expressions.

To anticipate potential roles for *TCP* genes in hormonal regulation, we examined the expression profile of *TCP* genes in response to a range of phytohormones, such as ABA, ethylene, GA, auxin, and MeJA (Figure 8B, Appendix A). In reaction to ABA’s hormonal treatments, *PeTCP17*/*23*/*30*/*15*/*21* showed instantaneous induction. On ethylene treatments, *PeTCP15*/*17* exhibited preferential upregulation. Treatments with auxin resulted in elevated *PeTCP21*/*27*/*28*/*29*/*30* gene expression. In reaction to the GA treatment, *PeTCP15*/*17*/*22* showed induced expressions, while *PeTCP15*/*17*/*27* exhibited preferential upregulation in response to MeJA treatments.

*TCP* genes are crucial for protecting cells from stress-induced oxidative damage [37]. We investigated the expression profiles of *PeTCPs* under heat, cold (Figure 8C, Appendix A), salt, and drought stress (Figure 8D, Appendix A) conditions using transcriptome data. Many genes, including *PeTCP15*/*16*/*17*/*19*, showed increased gene expressions and responded immediately to the cold treatments. Similarly, *PeTCP16*/*17*/*20*/*25* exhibited quick accumulation of mRNA transcripts in a short period after being rapidly induced under heat stress, indicating their speedy response to heat stress conditions. Furthermore, in response to salt stress, elevated gene expression was observed in CIN-type *PeTCP17/19/29/30* and PCF-type *PeTCP11/15/16/25*. The gene expression of *PeTCP15*/*16*/*17*/*19*//*25* increased in response to the drought stress.

GO and KEGG enrichment analyses of passion fruit were carried out for 30 *TCP* genes (Figure 9). The *PeTCPs* were majorly enriched with GO terms such as regulation of transcription (*PeTCP7*/*25*), response to lipids (*PeTCP3*) and hormones (*PeTCP7*), inflorescence development (*PeTCP7*), defense responses (*PeTCP6/7*), response to stimulus (*PeTCP27*), and amino acid binding (*PeTCP16*), while only a single gene, *PeTCP25*, was significantly enriched for KEGG pathways for plant circadian rhythms.

Given the considerable impact of low temperatures on passion fruit cultivation [38], we selected eight notable *PeTCPs* for qRT-PCR analysis based on their significantly varied expression in response to cold stress at 6 °C imitating the local winter climatic conditions of Guangxi Province, China [16]. In conformation to the transcriptomics data, CIN-type *PeTCP19*/*17*/*23* and PCF-type *PeTCP16/15* expressions were significantly increased by application of cold stress treatment at 6h intervals, declined sharply at 12 h intervals, and steadied at 24 h (Figure 10), suggesting expressional induction of these genes. Contrarily, PCF-type *PeTCP25*/*1* and CYC-type *PeTCP11* exhibited decreased expressions with subsequent cold treatments compared to control (Figure 10). Interestingly, among treatment intervals, the 6 h interval was the most influential. These results indicate that passion fruit’s *TCP* genes are positively or negatively induced by cold stress.

### 3.8. Subcellular Localization of PeTCP Proteins

The nuclear localization of transcription factors is widely recognized as crucial for regulating the transcription of target genes by binding to specific *cis*-regulatory elements in their promoters [39]. Earlier studies in grapevines [39] and strawberries [40] suggested TCP proteins are majorly localized to the nucleus. In this research, most PeTCP proteins were forecasted to be localized in the nucleus using Euk-mPLoc (Table 1). To study PeTCP locations within cells, two cloned *PeTCP* genes (*PeTCP1-GFP* and *PeTCP19-GFP*) were inserted into the pCAMBIA1300-GFP vector harboring the CaMV 35S promoter. The fusion constructs were then introduced into the epidermal cells of *Nicotiana benthamiana*, and the green fluorescence emitted by the PeTCP1-GFP and PeTCP19-GFP fusion proteins was specifically found within the nuclei, as validated by a mCherry-labeled nuclear marker (H2B-mCherry) (Figure 11). Our findings suggest that PeTCP1 and PeTCP19 proteins localize within the cell nucleus, aligning with predicted outcomes.

## 4. Discussion

Tropical fruit crops like passion fruit have significant agricultural, commercial, and ornamental value, however, environmental conditions have a big impact on the fruit’s growth and development. TCP transcription factors play pivotal roles in growth and development and coping with biotic and abiotic stresses.

In the current study, a total of 30 members of the TCP transcription factor gene family in the passion fruit genome were identified. Using homologous genes from *Arabidopsis*, phylogenetic analysis exhibited *TCP* gene family divergence into two classes and three subfamilies (Figure 1 and Figure 2). The results of the phylogenetic analysis and classification were independently supported by domain architecture, and gene exon/intron structure analysis indicated that closely related gene members typically exhibit similar structural characteristics (Figure 3), as observed in other plants like rice [41] and rye [25]. We predicted that most of the TCP proteins were located in the nucleus except a few in the cytoplasm (Table 1). There is a chance that some TCP proteins might be transiently located in the cytoplasm, as suggested by subcellular localization assays of strawberry FvTCP7 and FvTCP17. These proteins exhibited a punctated pattern of GFP signals in *Arabidopsis* mesophyll protoplasts [40]. Furthermore, the results of homologous protein modeling of the representative proteins of each gene subfamily exhibited divergent 3D structures and distinctive features associated with their varied functions (Figure 4). The structural diversity of these proteins may contribute to the functional diversity of *TCP* gene subfamilies.

Passion fruit had a slightly higher number of TCP proteins than *Arabidopsis* with 24 protein members. Among the 30 total *TCP* genes in passion fruit, five pairs of segmental duplication were identified, largely responsible for the expansion of the *TCP* family. The Ka/Ks ratios of all the duplicated gene pairs were less than 1 (Appendix A), suggesting that these pairs have undergone negative or purifying selection over their evolutionary history. Except for the lowly expressed duplicated pairs *PeTCP18*/*24* and *12/26*, the other members of duplicated gene pairs exhibited opposite or varied gene expressions, suggesting these genes following duplication might have experienced subfunctionalization or neofunctionalization. In contrast to model laboratory species such as *Arabidopsis*, passion fruit had to adapt to a greater variety of abiotic stressors to flourish during its growth and development. The expansion of *PeTCPs* and their diverse roles may enhance plants’ resilience to various adverse conditions, thereby improving passion fruit’s adaptability to changing environments.

In the meantime, four genes (*PeTCP19/20/29/30*) potentially targeting miRNA319 in passion fruit were identified through bioinformatics analysis (Figure 7). Interestingly, all four genes belonged to the CIN subfamily and showed similar expression profiles (Figure 8C). miR319 is one of the primitive and the most evolutionarily conserved miRNAs in plants [42]. Growing data indicate that miR319-regulated *TCP* (*MRTCP*) genes play a major role in the development of plants and stress responses [15]. Interestingly, miR319 appears to have conserved roles in modulating stress responses in passion fruit, as evidenced by differential expression of all three targeted genes (*PeTCP19*/*29*/*30*) under stress conditions (Figure 8C and Figure 9). These findings predict that miR319 may influence the transcriptional levels of *TCP* genes in passion fruit, affecting growth, development, and stress responses. However, further experimental validation of the predicted miRNA319-*PeTCP* module is required relating to the cold stress tolerance.

Although *TCP* genes’ involvement in growth/development and stress management has been reported in other plant species, studies of the *TCP* gene family in passion fruit are missing. The functional variety of the *TCP* genes was identified by analysis of the gene expression of the passion fruit. Some genes showed tissue-specific expression. For instance, *PeTCP17/23* exhibited obvious high expressions in petals and stamen tissues, whereas *PeTCP29* showed preferential upregulation in leaf and root tissues (Figure 8A). Additionally, *PeTCP15* exhibited obvious high expressions in petals and immature fruit tissues. Transgenic expression of *CmTPC14* in *Arabidopsis*, which is a transcription factor of the chrysanthemum *TCP-P* gene family, resulted in postponing senescence and reduction of organ size [43]. TCP TF RETARDED PALEA1 (REP1) of *Oryza sativa* has interactions with protein AT-hook DEPRESSED PALEA1 (DP1), similar to maize, suggesting an evolutionarily preserved pathway of inflorescence and flower development in the gramineae family [44].

Members of the *TCP* gene family are known to be crucial components of hormonal signaling networks in various plant species [45]. In response to ABA treatments, *PeTCP15*, *PeTCP17*, *PeTCP21*, and *PeTCP30* exhibited immediate induction. *PeTCP15*/*17* were upregulated upon ethylene treatments as well. Auxin treatments led to increased gene expression of *PeTCP21*/*27*/*28*/*29*/*30* (Figure 8B). Furthermore, the snapdragon CIN gene is linked to the regulation of genes involved in cytokinin and auxin signaling pathways along with lateral organ formation [46]. *PeTCP15*/*22*/*17* exhibited expressional induction responding to the GA treatment, meanwhile, *PeTCP15*/*17*/*27* showed preferential upregulation in response to methyl jasmonate (MeJA) treatments (Figure 8B). The altered gene expression levels measured during MeJA treatment in *Senna tora* suggested that *StTCP11* and *StTCP4.1* may be implicated in jasmonic acid (JA) response signaling [47].

Passion fruit’s growth and development are extremely susceptible to fluctuations in the climate. The expression of some gene members, such as *PeTCP15*/*16*/*17*/*19*, increased rapidly with the cold treatments (Figure 8C). It is well documented that the miR319-*TCP* module regulates cold stress in rice [17,18,41], sugarcane [19], and cassava [20]. Similarly, *PeTCP16*/*17*/*20*/*25* were induced quickly under heat stress and showed high transcript accumulation in a short period (Figure 8C). Additionally, CIN-type genes *PeTCP17*/*19*/*29*/*30* and PCF-type *PeTCP11*/*15*/*16*/*25* exhibited enhanced mRNA transcripts in response to salt stress (Figure 8D). A comparative transcriptome investigation between salt-sensitive and salt-tolerant genotypes of common beans demonstrated salt-responsive expression patterns for *Pvul-TCP1*/*11*/*13*/*22*/*27*, which are targets of miR319 [48]. Furthermore, a recent study of the pecan (*Carya illinoensis*) *TCP* gene family reported that transgenic *Arabidopsis* CIN-type *CiTCP8* enhanced salt stress tolerance [49]. In response to the drought stress, *PeTCP17*/*19*/*15*/*16*/*25* showed increased gene expressions (Figure 8D). Drought treatments induced *ZmTCP42 and ZmTCP32* expressions, and overexpression of *ZmTCP42* in *Arabidopsis* resulted in enhanced drought tolerance [50], however, in another study overexpression of *ZmTCP14* showed the opposite results [23], confirming *TCP* roles in drought resistance. In *Allium senescens* during drought stress, eight members of the *TCP* gene family exhibited induced gene expressions, including the *AsTCP17* whose heterologous overexpression in *Arabidopsis* helped cope with drought stress [51]. Interestingly, all the *PeTCPs* belonging to the CYC subfamily exhibited low expression across all the samples, while four CIN subfamily genes, *PeTCP17*/*19*/*29*/*30*, exhibited strong differential expressions in reaction to the stress conditions (Figure 8). Furthermore, qRT-PCR expression validations of eight selective *PeTCPs* (Figure 10) suggested highly consistent results with RNA-seq data (Figure 8B). Finally, two representative proteins from each phylogenetic class (PeTCP1 from Class I and PeTCP19 from Class II) were confirmed to be localized to the nucleus (Figure 11), in conformation to the universal localization of transcription factors to the nucleus [38] and previous studies [39,40,49].

Some of these promising genes such as *PeTCP15*/*17*/*19*/*29*/*30* may have significant application potential for the genetic improvement of passion fruit with improved tolerance to abiotic stress because they respond to various stimuli. In their promoter regions, they had a variety of *cis*-regulatory elements (CREs) linked to hormonal responses (ABA responsiveness, MeJA responsiveness), growth (light-responsive element, meristem expression), and stress responses (anaerobic induction, low-temperature responsiveness) (Figure 6), which play a significant role in the transduction of biological information [52]. In rice, *TCP19* modulates drought-induced ABA signalling through its interaction with *OsABI4*, which encodes a TF implicated in the transmission of the ABA signaling [22]. AtTCP14 inhibits ABA signaling by interacting with the DNA binding with one finger (DOF6) TF, preventing the induction of additional ABA-responsive genes in *Arabidopsis* as well as the downstream ABA biosynthesis gene ABA deficient 1 (*ABA1*) [53]. In addition, the co-occurrence of several CREs in the promoter regions of *PeTCPs* may be intimately associated with the functions of these genes in the growth and development of passion fruit in response to various environmental modifications [54].

Recent developments in genetic modification, somatic hybridization, micropropagation, somatic embryogenesis, and cryopreservation have enabled transgenic improvements in the *Passiflora* genus [55,56]. Additionally, candidate passion fruit genes can be overexpressed in other organisms like *Arabidopsis*, tobacco, and yeast to study their role in plant development and stress tolerance/avoidance [38]. Therefore, further studies are required to help establish links between candidate *PeTCP* genes identified in this research and their phenotypic or stress responses.

## 5. Conclusions

This study is the first deciphering the phylogeny, gene structure, *cis*-regulatory elements, and expression of the *TCP* family, one of the important regulatory factors, in passion fruits. A total of 30 *PeTCPs* were identified distributed on nine chromosomes and can be classified into three subgroups. The motif composition and the 3D models of the TCP proteins further exhibited similarity at the basic helix I and loop regions and the variations at helix II. The differential expression patterns of *PeTCPs* across the tissues and organs were revealed, some of which were tissue-specific. In addition, transcriptome expression analysis of *PeTCPs* under cold, heat, salt, and PEG treatments highlighted their predominant roles under stress conditions. Finally, the subcellular localization of two PeTCP proteins was confirmed in the nucleus. Our results provide insights into functional analysis of *TCP* genes and a few of them (*PeTCP17*/*19*/*15*/*16*/*23* and *PeTCP1*/*11*/*25*) would be promising targets for the genetic improvement of cold stress tolerance of passion fruits.

## Figures and Tables

**Figure 1 plants-13-02568-f001:**
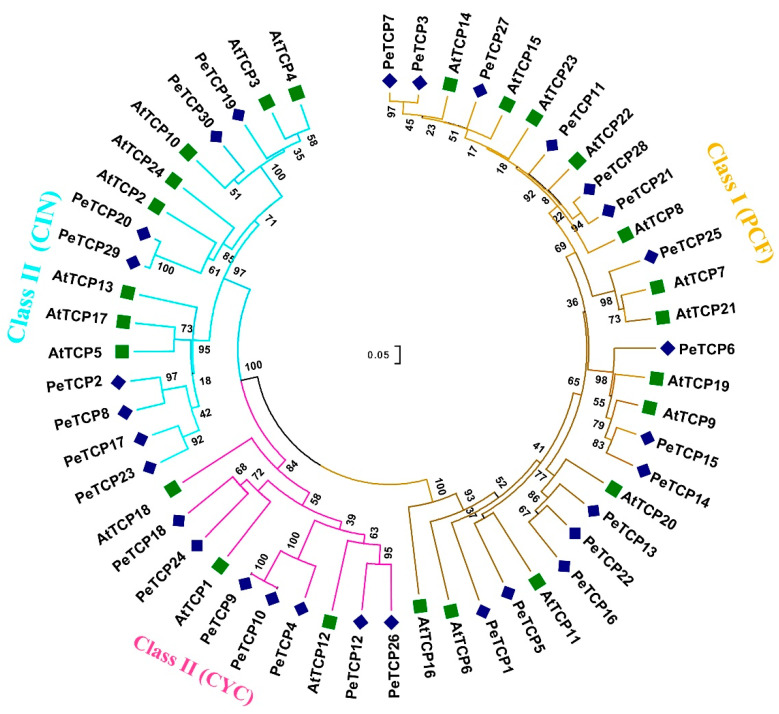
Phylogenetic analysis of *TCP* gene family proteins among passion fruit and *Arabidopsis.* The phylogenetic tree was built using MEGA-11 software employing the neighbor-joining tree method with 1000 bootstrap replicates. Passion fruit TCP proteins are designated by “Pe” while *Arabidopsis* proteins are depicted with the “At” prefix.

**Figure 2 plants-13-02568-f002:**
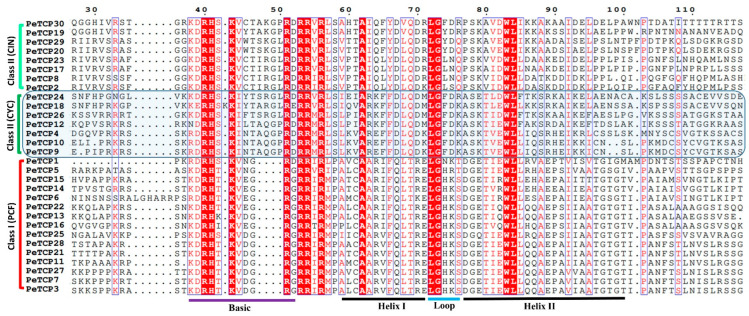
Multiple sequence alignment of passion fruit TCP proteins. Sequences were aligned in the MEGA software and sequence alignment was visualized in the ESPript 3.0 web tool. The residues highlighted in red vertical lines indicate completely conserved while those inside blue verticle lines are highly conserved amino acids.

**Figure 3 plants-13-02568-f003:**
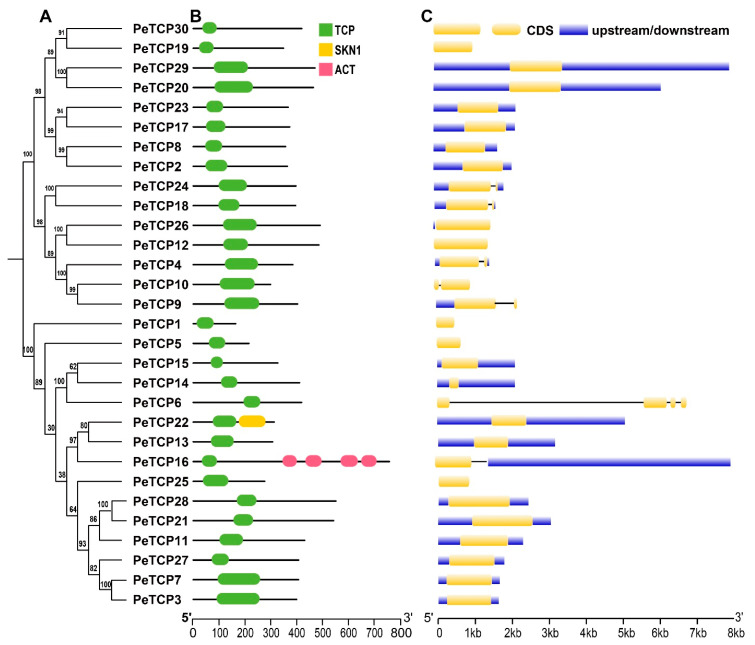
TCP protein domain composition and gene structure organization. (**A**) The phylogenetic tree was generated in MEGA-X software. (**B**) Domain attributes were downloaded from the NCBI batch CD server. TCP, SKN1, and ACT domains are depicted in green, yellow, and pink, respectively. (**C**) The gene coordinate information was drawn through the TB tool. CDS, introns, and upstream/downstream regions of gene structure are shown in yellow, black, and blue, respectively.

**Figure 4 plants-13-02568-f004:**
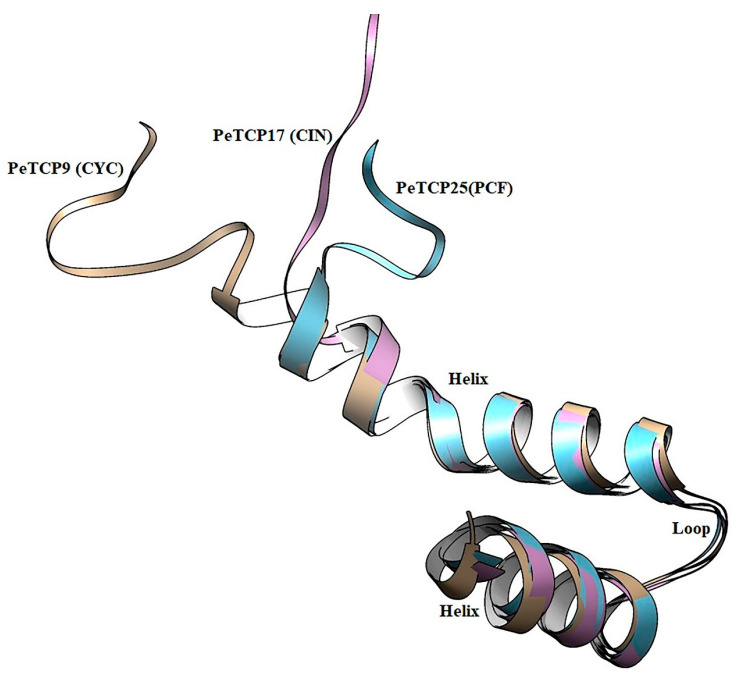
Structural modeling and superimposition of bHLH domains of three representative proteins from each subfamily of PeTCP. PeTCP9 depicted in brown, PeTCP17 shown in magenta, and PeTCP25 exhibited in blue represent CYC, CIN, and PCF subfamilies, respectively. Protein 3D modeling was performed using the SWISS-MODEL employing orthologous Arabidopsis TCP proteins as templates. Modeled proteins were visualized and structurally aligned using the Chimera software.

**Figure 5 plants-13-02568-f005:**
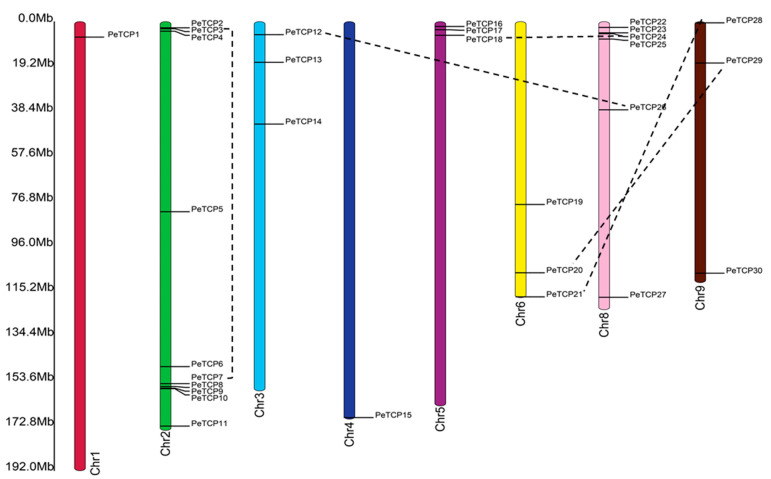
Distribution of 30 *TCP* genes in passion fruit genome. The chromosome structures and gene positions were depicted in TBtools employing the genomic information from the Passionfruit Genomics Network database. The scale on the left indicates the chromosome size in Mbps. The colored bars indicate chromosomes, while black dotted lines show the duplicated genes.

**Figure 6 plants-13-02568-f006:**
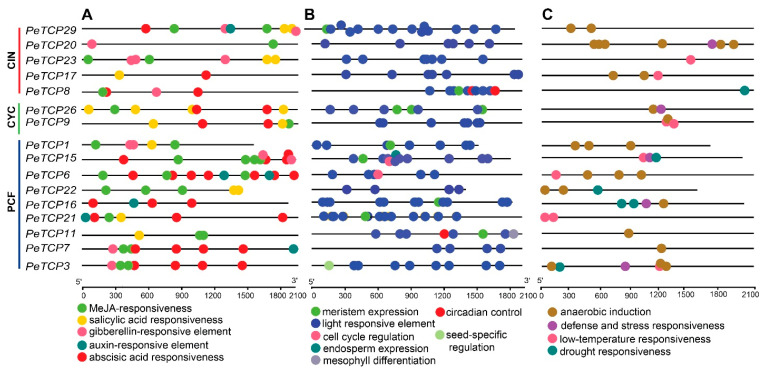
*Cis*-regulatory element (CRE) distribution in the predicted promoter regions of *PeTCPs*. (**A**) Hormonal responsiveness. (**B**) Growth and development related. (**C**) Stress responsiveness.

**Figure 7 plants-13-02568-f007:**
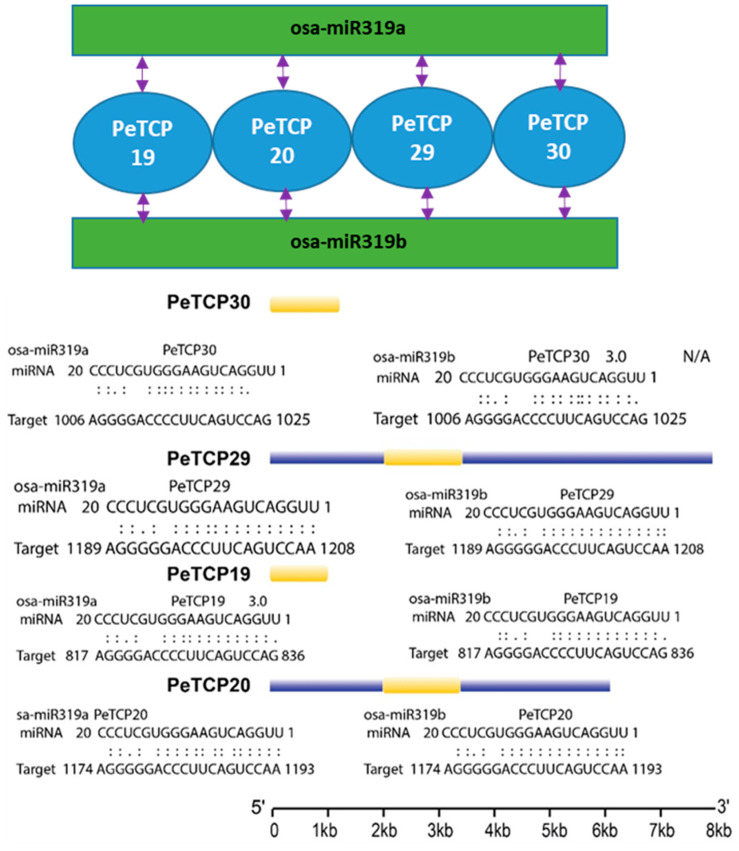
Predicted miRNA319a/b–*TCP*-binding modules.

**Figure 8 plants-13-02568-f008:**
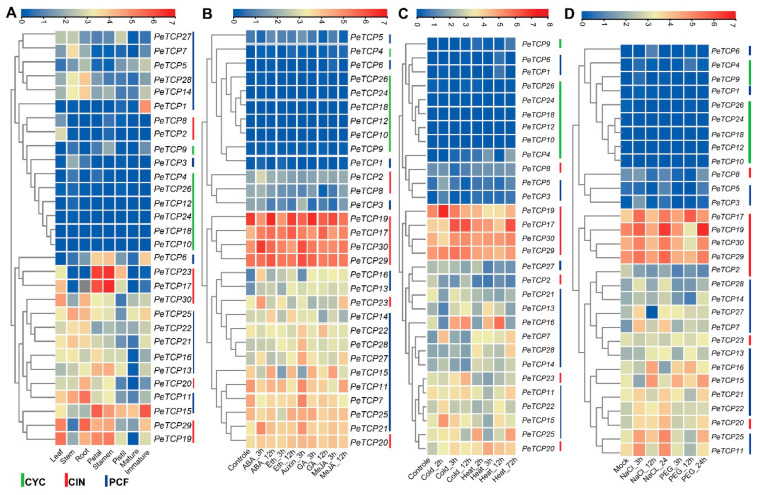
Transcriptome profiling of *TCP* gene expression levels. (**A**) Different growth stages and tissues. (**B**) Hormonal treatments. (**C**) Cold and heat stress conditions. (**D**) Salt and drought stress. The scale above each heatmap indicates the levels of relative gene expression, red, yellow, and blue correspond to high, medium, and low expressions, respectively. The vertical bars at the extreme right of each panel and below 8A indicate a phylogenetic subclass and green, red, and blue represent CYC, CIN, and PCF subclasses, respectively.

**Figure 9 plants-13-02568-f009:**
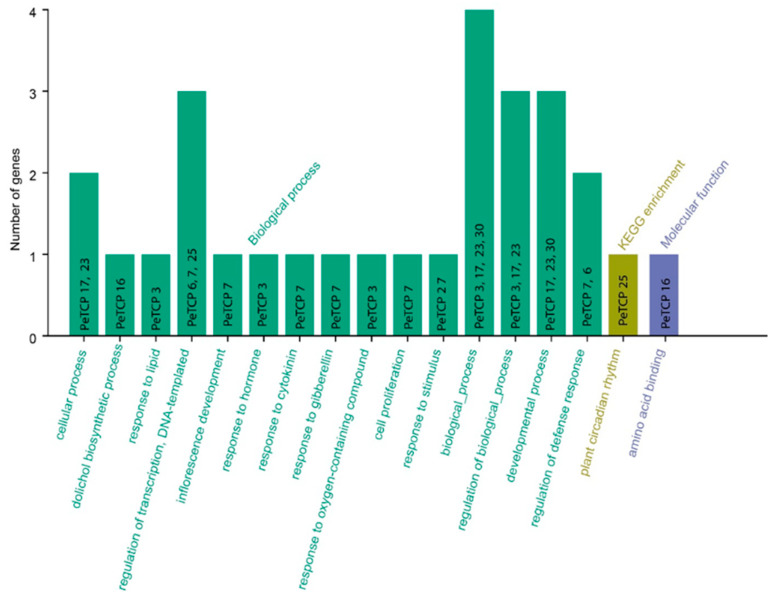
GO annotations and KEGG analysis of *TCP* gene family. The GO enrichment terms’ names are on the X axis while the number of genes belonging to each category is along the Y axis.

**Figure 10 plants-13-02568-f010:**
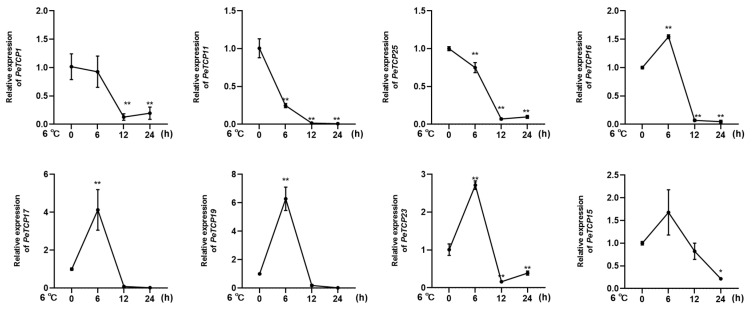
Expression profiles of 8 *TCP* genes (*PeTCP1, 11, 25, 16, 17, 19, 23, 15*) in response to the cold stress treatments. Three independent biological replicates’ standard deviations of means are represented by error bars. Significant variations of the transcript levels between treatments and blank control (0 h) are indicated by asterisks (* *p* < 0.05, ** *p* < 0.01).

**Figure 11 plants-13-02568-f011:**
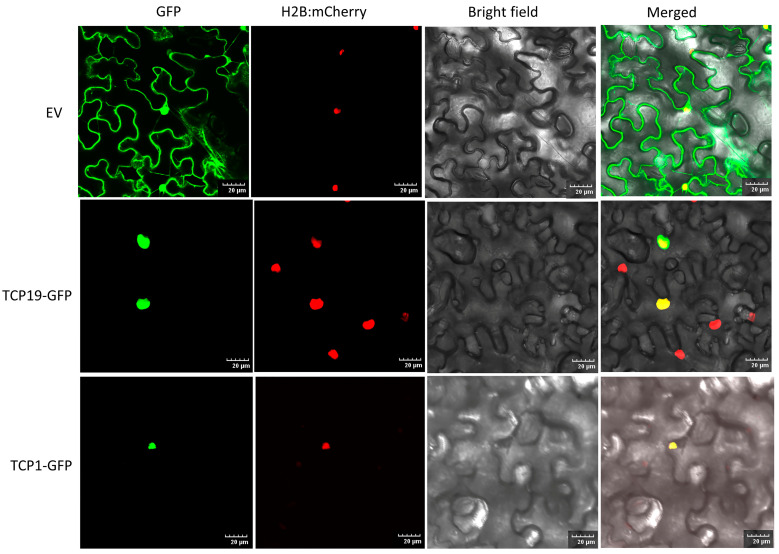
Subcellular localization of two representative TCP proteins. The top panels represent an empty vector (EV). The PeTCP19-GFP and PeTCP1-GFP fusion proteins were introduced into tobacco leaves for transient expression. After 60 h, confocal microscopy was used to observe the localization of these proteins. Nuclei were visualized using a co-transformed mCherry-labeled nuclear marker (H2B-mCherry). Scale bar, 20 μm.

**Table 1 plants-13-02568-t001:** Information of TCP family in passion fruit.

Gene Name	Gene ID	Chromosome	Size (aa)	MW (Da)	pI	A.I.	Stability	Gravy	Predicted Location
PeTCP1	Pe1g00801	LG01	164	17,985.36	6.1	58.35	Unstable	−0.438	Nucleus
PeTCP2	Pe2g00351	LG02	364	40,493.27	6.31	68.32	Unstable	−0.665	Nucleus
PeTCP3	Pe2g00400	LG02	400	42,807.17	7.91	62	Unstable	−0.65	Cytoplasm
PeTCP4	Pe2g00517	LG02	385	43,712.73	7.21	55.53	Unstable	−0.876	Nucleus
PeTCP5	Pe2g01516	LG02	215	23,168.9	6.71	65.95	Unstable	−0.641	Nucleus
PeTCP6	Pe2g02083	LG02	419	45,554.9	5.61	74.99	Unstable	−0.472	Membrane
PeTCP7	Pe2g02474	LG02	407	43,474.67	7.44	58.06	Unstable	−0.73	Nucleus
PeTCP8	Pe2g02555	LG02	357	39,723.46	6.51	64.23	Unstable	−0.622	Nucleus
PeTCP9	Pe2g02621	LG02	404	45,981.81	8.67	62.82	Unstable	−0.725	Nucleus
PeTCP10	Pe2g02632	LG02	299	34,140.84	9.86	58.8	Unstable	−0.894	Nucleus
PeTCP11	Pe2g03973	LG02	431	45,408.01	6.56	57.61	Unstable	−0.572	Nucleus
PeTCP12	Pe3g00812	LG03	486	54,179.6	6.01	54.09	Unstable	−0.9	Nucleus
PeTCP13	Pe3g01513	LG03	307	32,776.29	9.16	58.47	Unstable	−0.739	Cytoplasm
PeTCP14	Pe3g01959	LG03	411	42,952	6.37	67.98	Unstable	−0.43	Nucleus
PeTCP15	Pe4g04349	LG04	327	34,638.45	9.26	81.19	Unstable	−0.169	Membrane
PeTCP16	Pe5g00391	LG05	759	83,909.83	6.34	86.96	Unstable	−0.219	Chloroplast, membrane
PeTCP17	Pe5g00587	LG05	373	40,572.4	6.88	77.94	Unstable	−0.489	Nucleus
PeTCP18	Pe5g00665	LG05	396	44,016.51	9.29	64.29	Unstable	−0.685	Nucleus
PeTCP19	Pe6g00604	LG06	349	38,377.12	6.08	57.68	Unstable	−0.718	Nucleus
PeTCP20	Pe6g01133	LG06	465	50,516.54	8.7	60.86	Unstable	−0.808	Nucleus
PeTCP21	Pe6g02163	LG06	543	56,938.36	6.69	53.9	Unstable	−0.702	Nucleus
PeTCP22	Pe8g00389	LG08	313	33,367.08	7.99	62.75	Unstable	−0.718	Nucleus
PeTCP23	Pe8g00755	LG08	367	40,359.99	8.44	68.26	Unstable	−0.6	Cytoplasm
PeTCP24	Pe8g00826	LG08	397	44,330.83	8.46	70	Unstable	−0.626	Nucleus
PeTCP25	Pe8g01074	LG08	276	28,139.36	9.72	70.51	Unstable	0.353	Nucleus
PeTCP26	Pe8g02699	LG08	491	54,810.96	9.16	62	Unstable	−0.754	Cytoplasm
PeTCP27	Pe8g03645	LG08	408	44,573.37	6.7	62.5	Unstable	−0.688	Nucleus
PeTCP28	Pe9g00054	LG09	552	58,023.4	6.65	52.83	Unstable	−0.743	Nucleus
PeTCP29	Pe9g01413	LG09	471	51,180.1	8.84	59.68	Unstable	−0.856	Nucleus
PeTCP30	Pe9g02247	LG09	420	45,596.03	6.51	58.88	Unstable	−0.65	Nucleus

## Data Availability

Data are contained within the article and Appendix A.

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
