# Peer review of "Evolution and Functional Dynamics of TCP Transcription Factor Gene Family in Passion Fruit (Passiflora edulis)"

_plants, 2024, doi:10.3390/plants13182568_

Round 1
Reviewer 1 Report
Comments and Suggestions for Authors
Dear Authors
The results presented in the manuscript are devoted to the study of the TCP gene family and the proteins they encode, trans-factors, in an important tropical plant, passion fruit (Passiflora edulis). These genes are involved in the regulation of various physiological processes, including responses to damaging abiotic factors. This gene family is being intensively studied in a number of other objects, so the manuscript does not contain fundamentally new biological knowledge, although it makes a significant contribution to the understanding of the structure of TCP genes and the corresponding trans-factors in passion fruit plants. In my opinion, the study is carried out quite thoroughly, using primarily bioinformatics methods. The exception is the verification of the expression of eight genes of the above trans-factors under cold stress using qRT-PCR, as well as the assessment of the intracellular localization of two TCP proteins. The authors identified and analyzed 30 members of the TCP gene family in passion fruit, including the description of conserved domains and regulatory elements in the promoter regions of the genes, as well as their chromosomal localization. Of great interest are the expression profiles of the above genes depending on the stage of plant development and organ specificity, the effects of adverse factors of different nature and hormones. The expression of eight TCP genes under cold stress conditions was experimentally assessed and it was shown that OsmiRNA319 can use PeTCP as a potential target.
As in the overwhelming majority of similar studies, the authors do not explain on what basis they used this temperature (6 degrees above zero) as a cold effect, how plants respond to this factor from the point of view of a plant physiologist (does plant resistance to low positive temperatures change, does water status change, are ROS generated, are compatible osmolytes accumulated, etc.). The presented work is purely molecular in nature, so such requirements, I believe, can be ignored.
Kind regards
Author Response
Comment 1: As in the overwhelming majority of similar studies, the authors do not explain on what basis they used this temperature (6 degrees above zero) as a cold effect, how plants respond to this factor from the point of view of a plant physiologist (does plant resistance to low positive temperatures change, does water status change, are ROS generated, are compatible osmolytes accumulated, etc.). The presented work is purely molecular, so I believe such requirements can be ignored.
Response 1:
[Thank you for pointing this out. We agree with this comment. The reviewer raised an important point; however, we could not find research data on the physiological aspects related to low positive temperatures' effects on passionfruit. The general physiological changes plants undergo under cold stress are mentioned in the introduction section on page 2, paragraph 3, and lines 69-75. Regarding the choice of 6 degrees as a cold treatment, the revised manuscript changes can be found on page 14, paragraph 1, and lines 387-390.]
Reviewer 2 Report
Comments and Suggestions for Authors
I read this manuscript with great interest. It is well-written and well-structured and presents a comprehensive study on TCP transcription factors in passion fruit. The study is dynamic, using multiple methodologies, including bioinformatics, gene expression profiling, and functional assays. The findings and conclusions of this study are of great interest for future improvement of passion fruit in particular and crops in general. Here are some minor suggestions to improve the quality of the paper further.
Introduction: Please expand on passion fruit's economic and agricultural significance to provide a better context for the study.
Methods: I suggest providing additional details on the experimental design, particularly for the stress treatments and RNA-seq analysis. For better reproducibility, what were the growth conditions before the stress application?
Results: Improve the clarity and labeling of figures, ensuring that each figure is fully interpretable on its own.
Discussion: To strengthen the discussion and provide a broader perspective, compare TCP gene studies in other species.
Comments on the Quality of English Language- Perform a thorough review to correct any typographical errors and ensure consistency in formatting throughout the manuscript. For example, passion fruit is written as a single word (passionfruit) and two words (passion fruit) throughout the manuscript. Decide on one spelling
Author Response
Comments 1: [Introduction: Please expand on passion fruit's economic and agricultural significance to provide a better context for the study.]
Response 1: Thanks for your time and efforts in providing us with constructive and critical feedback on our article. We agree with the comments. [We have, accordingly, expanded the economic and agricultural significance of passion fruit to emphasize this point. This change can be found on pages 1, paragraph 2, and lines 42-45, and pages 1-2, paragraph 1, lines 47-50.]
Comments 2: [Methods: I suggest providing additional details on the experimental design, particularly for the stress treatments and RNA-seq analysis. For better reproducibility, what were the growth conditions before the stress application?]
Response 2: We agree with this comment. We have, accordingly, revised the method details to ensure reproducibility. [The revised manuscript modification can be found on page 4, paragraph 3, lines 176-177, and page 4, paragraph 5, lines 185-192, 195-196.]
Comments 3: [Results: Improve the clarity and labeling of figures, ensuring that each figure is fully interpretable on its own.]
Response 3: Agree. [The figure clarity and labeling have been improved as suggested.]
Comments 4: [Discussion: To strengthen the discussion and provide a broader perspective, compare TCP gene studies in other species.]
Response 4: We agree with this comment. We have, accordingly, discussed the TCP gene studies in other plant species. [The revised manuscript modification can be found on page 17, paragraph 4, lines 475-480, page 18, paragraph 1, lines 501-503, 508-510.]
Comments 5: [Comments on the Quality of English Language. Perform a thorough review to correct any typographical errors and ensure consistency in formatting throughout the manuscript. For example, passion fruit is written as a single word (passionfruit) and two words (passion fruit) throughout the manuscript. Decide on one spelling.]
Response 5: We agree with this comment. [We have, accordingly corrected the typographical error and ensured consistency in format throughout.]